# Avascular Necrosis of Femoral Head—Overview and Current State of the Art

**DOI:** 10.3390/ijerph19127348

**Published:** 2022-06-15

**Authors:** Wojciech Konarski, Tomasz Poboży, Andrzej Śliwczyński, Ireneusz Kotela, Jan Krakowiak, Martyna Hordowicz, Andrzej Kotela

**Affiliations:** 1Department of Orthopaedic Surgery, Ciechanów Hospital, 06-400 Ciechanów, Poland; tomasz.pobozy@onet.pl; 2Department of Social and Preventive Medicine, Social Medicine Institute, Medical University of Lodz, 90-647 Lodz, Poland; andrzej.sliwczynski.ahe@gmail.com (A.Ś.); jan.krakowiak@umed.lodz.pl (J.K.); 3Department of Orthopedic Surgery and Traumatology, Central Research Hospital of Ministry of Interior, Wołoska 137, 02-507 Warsaw, Poland; ikotela@op.pl; 4General Psychiatry Unit III, Dr Barbara Borzym’s Independent Public Regional Psychiatric Health Care Center, 26-600 Radom, Poland; m.hordowicz@gmail.com; 5Faculty of Medicine, Collegium Medicum, Cardinal Stefan Wyszynski University in Warsaw, Woycickiego 1/3, 01-938 Warsaw, Poland; andrzejkotela@gmail.com

**Keywords:** implants, avascular necrosis, femoral head, osteonecrosis

## Abstract

Avascular necrosis (AVN) of the femoral head is caused by disruption of the blood supply to the proximal femur. The alterations in the blood supply may occur following a traumatic event or result from a non-traumatic cause. Femoral neck fracture and hip dislocation and associated surgical procedures, corticosteroid therapy, and alcohol abuse frequently lead to AVN development. Type of fracture (displaced or undisplaced) and time between injury and surgery are the most critical factors in assessing the risk of developing AVN. Diagnosis of AVN can be established based on patients’ complaints, medical history, and radiographic findings. There is no consensus on the treatment of patients with AVN to date. Non-surgical methods are dedicated to patients in the early pre-collapse stages of the disease and consist of pharmacotherapy and physiotherapy. Surgery is recommended for patients with advanced disease.

## 1. Introduction

Avascular necrosis (AVN) of the femoral head is a type of aseptic osteonecrosis, which is caused disruption of the blood supply to the proximal femur, which results in osteocyte death. AVN may occur due to ischemia developing on a traumatic or non-traumatic background [1,2]. The most common etiological factors include treatment with corticosteroids, fractures, dislocation of the hip joint, and alcohol abuse [3]. It typically affects physically active people aged between 20 and 40 years. Between 10,000 and 20,000 new cases of AVN of the femoral head are diagnosed in the United States (US) each year [4]. In the United Kingdom, it is the third most common reason for total hip replacement (THR) in patients under 50 years of age [5].

Early diagnosis of AVN gives the physicians options beyond THA. If AVN of the femoral head is left untreated, it progresses to secondary hip arthritis in 70–80% of patients [3]. Surgical decompression of the femoral head reduces the likelihood of secondary surgery. Nevertheless, if the patient progresses to secondary hip arthritis, joint replacement surgery is indispensable [4,5,6].

This article offers an overview of the current literature regarding the pathophysiology and current concepts of AVN clinical management.

## 2. Etiology of AVN

The exact pathomechanism of AVN is often unclear. Each case is probably determined by different factors, including underlying conditions or medication that increase the likelihood of vessel obstruction, alteration of the osteocyte’s metabolism, and genetic factors [7].

Most of the blood supply to the femoral head in adolescents and adults comes from the medial and lateral circumflex branches of the profundal femoris artery, which derives from the femoral artery [8]. Obstruction of the subchondral microcirculation, particularly retinacular vessels, leads to bone necrosis. Bone cell necrosis is linked with a high risk of developing secondary osteoarthritis and restrictions in the hip range of motion through an accumulation of microfractures in the osteonecrosis area [2,8].

Common etiological factors of femoral head AVN are summarized in Table 1.

### 2.1. Non-Traumatic Causes of AVN

The most common non-traumatic causes are corticosteroid treatment and alcohol abuse. Corticosteroids alter adipocytes’ differentiation, increasing the size and number of adipocytes. This process leads to the intracellular accumulation of lipids. As a result of increased pressure inside the bone cells, vascular endothelial cells become damaged, leading to local coagulopathy, vascular thrombosis, and ischemia. A dose-dependent effect was observed in a meta-analysis performed by Mont et al. The incidence of AVN was 6.7% with a corticosteroid dose of >2 g prednisone equivalent per day. Every additional 10 mg/day increased the rate of AVN by 3.6%. Nonetheless, not all patients receiving corticosteroids develop AVN. It is proposed that other factors, including genetic polymorphisms and concomitant diseases, increase individuals’ vulnerability [7,10,11]. For instance, in patients treated with corticosteroids due to systemic lupus erythematosus, a higher risk of AVN was observed than in patients with other medical diagnoses [6]. Other drugs, such as antiretrovirals, are also linked to increased AVN risk [12,13].

Alcohol abuse is reported by 20–30% of the patients with AVN. The potential mechanism of induction of AVN is unknown. Alcohol may provoke osteocyte death through several pathways, e.g., by increasing intracellular deposition of triglycerides, which leads to pyknosis of osteocytes similarly to corticosteroids, and by decreasing osteogenesis through promoting stromal cell differentiation into adipocytes [14].

Other conditions related to the AVN include Gaucher’s disease. Patients who developed anemia in the course of that disease have increased chances of AVN development by 60% when compared with the non-anemic group [13]. In sickle cell hemoglobinopathy, ischemia is provoked by restrictions in local blood flow by abnormally adherent blood cells [15]. In the case of hematopoietic cell transplantation, concomitant immunosuppressive treatment with corticosteroids and graft versus host disease was proposed to induce microcirculation alterations and, consequently, AVN [16]. A less common cause is Legg–Calvé–Perthes disease, an idiopathic AVN in the pediatric population [17].

### 2.2. Traumatic Factors

Posttraumatic AVN occurs when the blood supply to the femoral head is disrupted due to a fracture or dislocation of the femoral head. In most cases, AVN is related to fractures in the sub-capital region of the femoral neck. Injury in this region disrupts the anastomosis between the lateral epiphyseal vessels, limiting the blood supply to the femoral head [4,18].

In the systematic review performed by Ghayoumi et al., the overall incidence of avascular necrosis in patients with displaced femoral neck fractures was 17.3% [19]. According to Slobogean et al., displaced fractures were associated with a statistically higher incidence of AVN than undisplaced fractures (14.7% vs. 6.4%) [20].

### 2.3. Avascular Necrosis following Stabilization of the Fractured Femoral Neck

Femoral neck fixation allows recovery of vascular supply to the femoral head and restores its functionality, thus avoiding joint replacement surgery. For internal fixation in non-displaced cases, most surgeons choose hip preserving techniques such as a dynamic hip screw (DHS) or multiple cannulated screws (MCS) [21]. DHS is a more invasive technique than MCS but achieves better anchorage with less rotation and cut-out [21].

Figure 1 and Figure 2 present AVN of the femoral head developed after surgical stabilization of an intertrochanteric fracture of the femoral bone.

### 2.4. Dynamic Hip Screw (DHS) Technique and the Risk of AVN

Many authors have described cases of AVN after a femoral neck fracture surgery. DHS was shown to be associated with a high risk of osteonecrosis. Some studies have demonstrated that it might be even higher than that with other hip-preserving techniques, such as cancellous screws. Schwartzman et al. observed AVN in 16% of 96 patients with sub-capital neck fractures treated with DHS, though not all other papers show a statistically significant difference between these two methods [21,22,23,24]. On the contrary, in a large randomized controlled trial, AVN was more common in patients treated with DHS than in those treated with cancellous screws (9% vs. 5%, HR 1.91, 1.06–3.44; *p* = 0.03) [25].

Using large implants for fixation (such as those used in DHS) of femoral neck fracture is associated with a disruption in blood supply to the femoral head. This was confirmed using bone scintigraphy in 104 patients with femoral neck fractures, indicating reduced vascularity in patients treated with DHS compared with those treated with cancellous screws (35% vs. 11%, *p* < 0.01) [26]. Other potential complications of surgical treatment of femoral head fractures include delayed union, nonunion, infection, and angular or rotational malalignment [27].

Some studies have shown promising results regarding AVN risk reduction following the modified DHS technique. Elgeidi et al. described the outcomes of using DHS and autogenous fibular strut graft for fixation of a femoral neck fracture with posterior comminution. Interestingly, the authors did not observe any case of AVN. This could be attributed to the fact that fibular graft has an osteoconductive and osteoinductive potential and acts as a biologically compatible implant allowing revascularization [28].

### 2.5. AVN following Fixation with Cannulated Screws

Some studies demonstrate that the use of cannulated screws may also be linked to a high rate of complications. Duckworth et al. observed a 32% rate of complications after fixation of intracapsular fracture of the femoral head by cannulated screws. The second most common reason for failure was AVN, observed in 11.5% of the patients with a mean time to treatment failure of 19.8 months [29]. Other authors reported similar rates of AVN after cannulated fixation [30,31,32].

Modification of the cannulated screw method was developed to decrease the odds of AVN. Li et al. compared the occurrence of AVN after fixation of the femoral neck using three cannulated screws with and without deep circumflex iliac artery bone grafting (DCIABG). The rates of AVN were significantly lower in the group of patients treated with DCIABG compared to classic fixation (9.7% vs. 26.8%, *p* < 0.001) [33].


*New hip-preserving techniques under investigation*


Preliminary results of using Targon femoral neck (TFN) implants are promising in terms of reduction in ANV risk following femoral neck fracture fixation. The TFN implant is a locking plate system with telescoping sliding screws. Depending on the fracture type, internal fixation of femoral neck fracture with a TFN implant may lower the risk of AVN to as low as 3% [34,35,36]. Though data on the incidence of AVN after short intramedullary femoral nailing are limited, it seems to be promising in terms of AVN risk reduction. Chen et al. reported a 1.2% incidence of AVN among patients with unstable, intertrochanteric fractures treated with Asian Pacific gamma-nail [37].

### 2.6. Other Factors to Consider When Assessing the Risk of AVN Following Femoral Neck Fixation

Estimation of the risk of AVN may help in optimizing treatment plans in patients after femoral neck fracture. Factors such as type of fracture, Garden classification, preoperative traction, and the time interval between injury and surgery seem to be crucial in terms of the risk of AVN development [38]. Displaced fractures in elderly patients are usually unsuitable for hip preserving techniques and require hemiarthroplasty or THA; the risk of AVN is higher in displaced fractures than in undisplaced ones [39]. In another study by Loizou et al., most patients with hip fractures underwent internal fixation using three cannulated screws. The overall incidence of AVN was 6.6%, and the complication was more common in patients with displaced fractures than in those with undisplaced ones (9.5% vs. 4.0%, *p* = 0.0004). Of note, contrary to other reports, AVN was more common in women than in men regardless of the type of fracture (displaced vs. undisplaced) [40].

The time between surgery and injury appears to be crucial in assessing the risk of bone ischemia—a time interval between injury and surgery greater than 24 h was associated with an increased risk [39]. This was confirmed by Migliorini et al. In that systematic review, data from 6112 patients with osteonecrosis of the femoral head who underwent hip-preserving procedures were retrieved to identify prognostic factors for failure of initial surgical management and conversion to total hip arthroplasties (THAs). Longer duration of symptoms and higher VAS score before treatment resulted in reduced time to treatment failure. In addition, a poor hip function was related to an increased rate of treatment failure and conversion to joint replacement. Female gender was a protective factor against treatment failure and conversion to THA and increased time to that procedure. Other analyzed factors such as body mass index, the cause of AVN, and time from surgery to full weight-bearing did not significantly affect the treatment outcome [41].

### 2.7. AVN Following Hip Fracture in the Pediatric Population

AVN cases in children require separate discussion. In the pediatric population, the incidence of AVN may be higher than that in adults. In the study of Bali et al. 36 children with a mean age of 10 years sustained femoral neck fractures. Patients were treated conservatively, with open reduction and internal fixation or closed reduction and internal fixation. Of 13 cases treated conservatively, 8 had lost reduction and required surgical intervention. The rate of AVN was 19.4%, which is higher than the average rate observed in adults. A plausible explanation is that the adult hip has intraosseous blood vessels that supply the femoral head, and these blood vessels are absent in children as they cannot cross the physis, which remains open at a young age. Therefore, the blood supply in children may be disrupted more easily after a hip fracture. In the literature, the rates of AVN following acute trauma in children ranged from 17% to 47% [42].

## 3. Clinical and Radiographic Examination of the AVN

The diagnosis of AVN is mainly based on both clinical and radiographic findings. Typical clinical presentation includes increasing pain, stiffness, and crepitus, usually proceeded by a period of minimal symptoms. During the physical examination, patients typically complain of a limited range of motion at the hip and the presence of pain, particularly with a forced internal rotation [4]. Early identification of the disease provides better outcomes. Many imaging techniques were found helpful in detecting bone necrosis signs, including X-ray, magnetic resonance imaging (MRI), computed tomography (CT), and radionuclide examinations. Imaging evaluation of AVN should begin with radiography, a non-expensive and widely available technique. Classic radiography may show subchondral radiolucency, called the “crescent sign”, indicating subchondral collapse [9]. CT and X-ray are less sensitive than MRI and show the necrotic changes during later stages of AVN. Nonetheless, signs of AVN are often apparent enough not to warrant additional radiologic evaluation [43]. Typical findings on MRI in a patient with an AVN are pictured in Figure 3.

MRI is the gold standard for osteonecrosis diagnosis and allows differentiating AVN from other diagnoses that may mimic it, such as bone bruises or transitioned osteopenia [1,9]. MRI allows for early AVN diagnosis and may help identify patients at risk of femoral head fracture. Identification of bone marrow edema in the proximal femur and joint effusion are critical prognostic factors [3]. T1-weighted images show a limited subchondral linear-shaped low signal intensity, while T2 demonstrates a double-line sign [44]. However, MRI cannot be used after fracture fixation with metallic implants, limiting its utility, especially in patients who develop bone ischemia following surgical procedure [18].

Fan et al. compared single-photon emission computerized tomography and computerized tomography (SPECT/CT) to determine the risk of bone necrosis in patients following femoral neck fracture. The study results revealed that SPECT is most useful for determining the prognosis of AVN in patients aged >58 years and with displaced fractures [45]. Diagnostic methods based on nuclear medicine, such as positron emission tomography (PET) or technetium bone scans, may also be used to detect the early stages of AVN and help predict the disease progression [1,46].

Although the patient’s medical history, clinical features, and radiographic examination might indicate AVN, the clinician should include other clinical entities in the differential diagnosis. These are summarized in Table 2.

The Steinberg University of Pennsylvania system is the classification most used in AVN (Table 3). This system includes six stages with the assessment of involvement within each stage. The classification allows for distinguishing between mild (<15% radiographic involvement of the femoral head), moderate (15–30% involvement of the femoral head), and severe (>30% involvement of the femoral head) stages [9,47]. The classification is presented in Table 3.

## 4. Management

### 4.1. Non-Surgical Management

Conservative treatment of AVN aims to improve hip function, prevent the femoral head from collapsing, provide pain relief, and delay necrotic changes [3,48]. Nonoperative management is mainly reserved for the early stages of disease in patients without a history of trauma. Usually, the imaging findings correspond to stages 0 and 1 on the Steinberg scale [1]. Restriction in weight-bearing using a cane, crutches, or a walker is one of the ways to delay disease progression. However, some papers indicate that reducing joint reactive forces does not slow disease progression [49].

### 4.2. Pharmacological Treatment

Multiple pharmacological agents were proposed as a treatment for AVN. These include anticoagulants, statins, vasodilators, bisphosphonates, and other agents currently under investigation [48]. Such treatment is mostly used at the early stages of the disease. Their effectiveness, however, is limited, and there are no clear recommendations for their use in AVN due to paucity of evidence. Many patients, after pharmacological treatment, eventually undergo surgery.

#### 4.2.1. Bisphosphonates

Bisphosphonates are recommended in the early stages of AVN. They act by inhibiting osteoclastic activity and reducing bone turnover, thus preventing woven bone formation [3]. In a randomized controlled trial, the efficacy of alendronate and placebo was compared in patients with non-traumatic AVN at Steinberg stages II–III. Patients in the drug arm experienced 2 collapses out of assessed 29 femoral heads, while 19/25 assessed femoral heads collapsed in the placebo arm [50]. However, another prospective, randomized, placebo-controlled study by Chen et al. did not confirm these findings. There were no significant differences in radiographic outcomes, prevention of THA, and improvement of quality of life between the placebo and treatment arm [51]. The results of available studies are therefore inconclusive. Some of them have limitations in their methodology, including the lack of a control group. The paucity of available evidence does not allow forming guidelines informing the dose and duration of bisphosphonate therapy.

#### 4.2.2. Statins

Therapy with statins may inhibit corticosteroid-induced adipogenesis and osteonecrosis of the femoral head. Nonetheless, similarly to bisphosphonate therapy, there are no guidelines on statin use. The results of Ajmal et al. indicate no difference in the occurrence of osteonecrosis between patients on corticosteroids receiving or not receiving statins [52]. On the contrary, Prichett et al. observed a significant reduction in AVN rate in patients on steroids and receiving statins [53].

#### 4.2.3. Vasodilators

The beneficial effect of a vasodilator iloprost on radiographic and clinical outcomes in patients with early stages of AVN was reported. Claßen et al. investigated the effect of iloprost in 108 patients with osteonecrosis; the median follow-up of patients was 49.7 months. Most of the patients (74.8%) noted an improvement in subjective complaints and a decrease evaluated by the visual analog scale. However, patients with a lower stage of disease had better outcomes [54].

Some authors suggest that enoxaparin may delay the progression of osteonecrosis if therapy is implemented in the early stages of the disease [55], but data on its effectiveness remain limited.

#### 4.2.4. Other Therapies

Different shockwave devices were studied in AVN treatment. Several studies involving extracorporeal shockwave therapy (ESWT) in AVN with promising results have been published [56]. The main effect observed was a decrease in pain; some patients had a complete regression of MRI changes. ESWT’s proposed mechanism of action is a stimulation of osteoblastic activity, which results in increased density of the bone in the pelvic area. Russo et al. stated that ESWT efficacy is more significant in the early stages of the disease and that ESWT is more effective than core decompression and grafting [57].

Hyperbaric oxygen therapy increases extracellular oxygen concentration and reduces intraosseous hypertension and bone edema. Nonetheless, due to small populations in clinical trials and limitations in their methodology, the efficacy of hyperbaric oxygen therapy has to be confirmed in large randomized controlled trials [58].

### 4.3. Surgical Treatment

Surgical treatment of AVN includes joint preserving procedures. These are mostly reserved for young patients in the pre-collapse stage of the disease. THA is indicated for patients with advanced disease.

Core decompression (CD) is the most common procedure performed in the early stages of AVN. The principle of this method is to reduce the intraosseous pressure and restore circulation in the femoral head. The procedure is based on drilling holes into the femoral head, which relieves internal pressure and creates space for new blood vessels [59]. CD is recommended as a first-line treatment for patients with early disease. The method is cost-effective and provides excellent results in long-term follow-up [1,49]. The technique improved over time, and now, the recommended way is multiple drilling [60,61,62].

Nonvascularized bone grafts derived from different body parts (e.g., tibial autograft, fibular autograft, or allograft) are used to fill the necrotic area in the femoral head. The procedure is most often used at the early stages of the disease after CD fails. There are three main techniques of nonvascularized bone grafting: trapdoor, lightbulb, and Phemister [63]. Many studies have reported excellent outcomes in patients after nonvascularized bone grafting [64,65,66,67,68].

Vascularized grafting improves subchondral architecture and also restores circulation in the damaged area of the femoral head. This technique uses part of the fibular bone with a nutrient artery. The graft is inserted into the decompression core with simultaneous anastomosis of the graft to the lateral circumflex femoral artery. This technique is recommended to treat patients with articular collapse <3 mm and <50% involvement of the femoral head [1].

Osteotomies are intended to redistribute the necrotic femoral head tissue away from the load-bearing area and replace it with the portion of the healthy tissue. However, despite protecting the weight-bearing region from collapse, the transferred necrotic area may induce joint instability and arthritic changes [4].

Cellular therapies are mainly based on mesenchymal stem cells (MSCs) derived from bone marrow, adipose tissue, or umbilical cord. MSCs initiate the revascularization process and regenerate bone tissue [69,70]. Li et al. treated 49 hips in the early stage of AVN with bone marrow MSCs combined with pravastatin. Results show improvement in hip function and pain. New vasculature was observed in 21 hips after six months from the treatment. While promising results of MSC therapy are observed, more controlled studies should be performed to make recommendations for its use in routine practice [71].

Total hip arthroplasty (THA) should be performed in patients with a significant femoral head collapse, loss of hip function, and severe pain. The procedure involves the removal of the ball and socket of the hip and replacement with an artificial implant (Figure 4). THA is a suboptimal choice for young patients due to activity restriction and possible future revision of the implant [63]. Most patients, however, have good outcomes after THA, particularly pain relief and restoring hip function [48,72].

## 5. Conclusions

AVN of the femoral head is a common cause of disability in patients aged between 20 and 40 years. AVN may be a complication of surgical treatment of femoral head fractures. AVN risk following such procedures is diverse and depends on several factors: the type of internal fixation, type of fracture, Garden classification, preoperative traction, and the time interval between injury and surgery. Untreated AVN leads to secondary hip arthritis requiring hip arthroplasty. Early identification of AVN offers physicians time to make a relevant treatment decision. Pre-collapse treatment is crucial to obtaining successful outcomes in patients. MRI is the gold standard as it allows for identifying osteonecrosis in its early stages.

Nevertheless, from a practical standpoint, imaging evaluation of AVN should begin with traditional radiography because it is a non-expensive and widely available technique. A trial of non-operative management should be performed in patients with early-stage disease, while surgical treatment is routinely used in more advanced stages. Though data on newer therapies are emerging, there is still little evidence to create precise recommendations regarding most treatment methods for patients with AVN.

The primary aim of this review was to provide a concise and practical overview of current knowledge of the pathophysiology of AVN, as well as clinical aspects, such as diagnosis, staging, and therapeutic options. Selected new AVN management methods, which are likely to be included in the routine clinical practice, were also briefly described. As clinicians, we are aware that such comprehensive overviews prepared by practicing specialists, though subject to the flows of narrative review, are of value to other clinicians, as they synthesize the evidence described in multiple systematic reviews and clinical studies, which, by definition, focus on specific populations or types of management. As authors, we hope that this brief overview of clinically essential aspects of AVN of the femoral head and its management would help inform practicing specialists. At the same time, we understand that narrative synthesis might not be appealing to, e.g., researchers and academics due to limitations in the methodology of such reviews.

## Figures and Tables

**Figure 1 ijerph-19-07348-f001:**
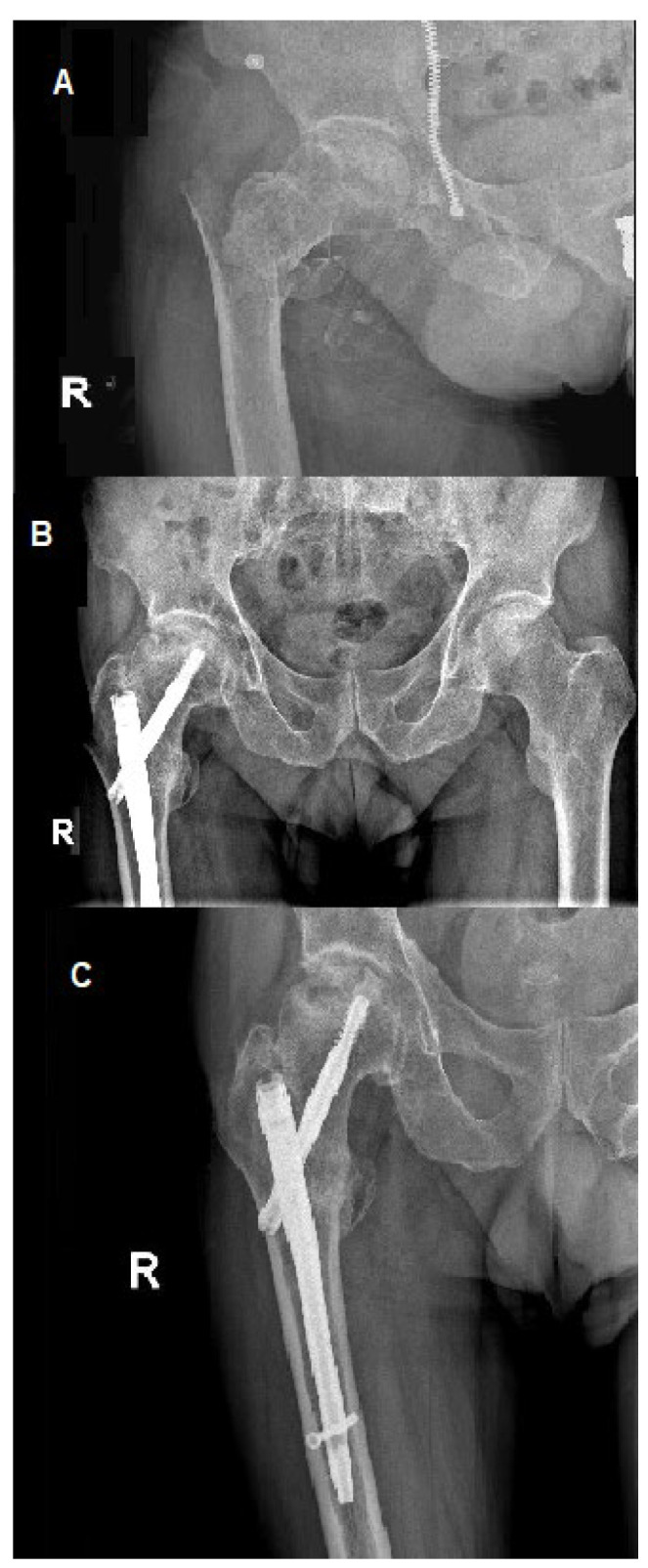
Avascular necrosis of right femoral head following stabilization of intertrochanteric fracture. (**A**) Posteroanterior view of the fracture; posteroanterior (**B**) view after 12 weeks of procedure; (**C**) view at a follow-up visit 20 weeks after the procedure, prominent necrosis of right femoral head; R—right.

**Figure 2 ijerph-19-07348-f002:**
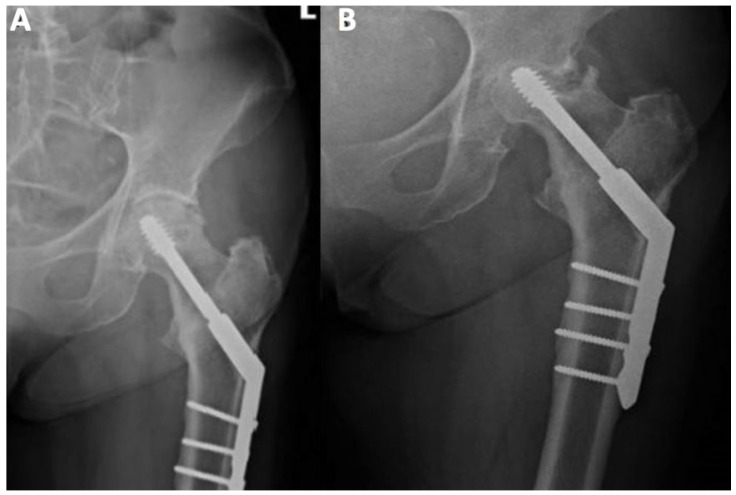
Posteroanterior view of internal fixation of left femoral neck fracture using the dynamic hip screw. (**A**) Posteroanterior view directly after the procedure; (**B**) posteroanterior view during control 15 weeks after the procedure, prominent necrosis of left femoral head.

**Figure 3 ijerph-19-07348-f003:**
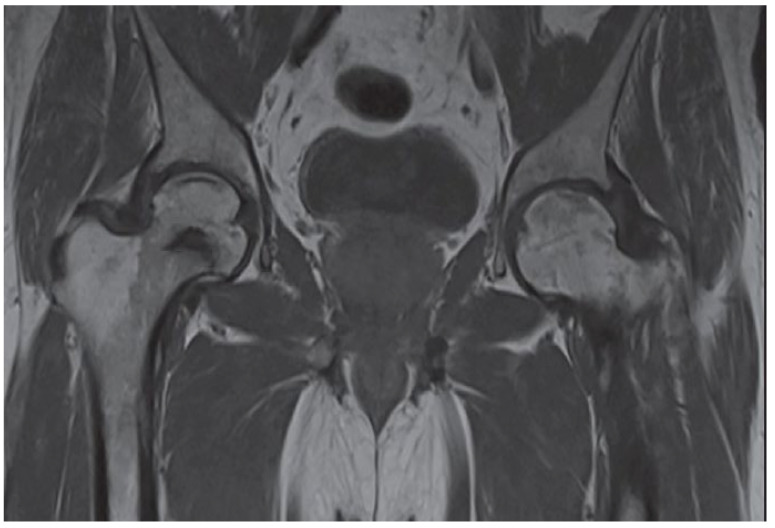
The posteroanterior view shows a right (R) AVN of the femoral head (T1-weighted).

**Figure 4 ijerph-19-07348-f004:**
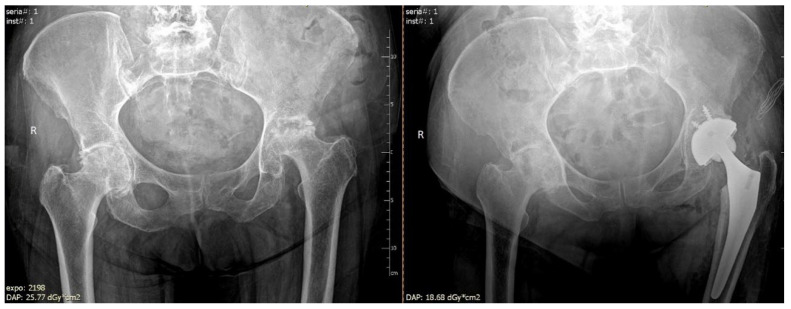
Arthroplasty of the left hip due to avascular necrosis of the femoral head.

**Table 1 ijerph-19-07348-t001:** Traumatic and non-traumatic causes of avascular necrosis of the femoral head [1,9].

Traumatic Causes	Non-Traumatic Causes
Fracture of the femoral head Hip dislocation	Corticosteroids
	Alcohol abuse
	Lupus erythematosus
	Gaucher’s disease
	Sickle cell disease and other hemoglobinopathies
	Bone marrow transplant
	Antiretroviral treatment
	Legg–Calvé–Perthes disease (in children)

**Table 2 ijerph-19-07348-t002:** Differential diagnosis of AVN of the femoral head.

Clinical Entities That Should Be Included in the Differential Diagnosis of AVN of the Femoral Head [44]
hip osteoarthritis
osteoarthritis secondary to acetabular dysplasia
ankylosing spondylitis of hip joint
transient osteoporosis or bone marrow edema
chondroblastoma of the femoral head
incomplete fracture in subchondral bone
pigmented villonodular synovitis
synovial herniation
femoroacetabular impingement syndrome
bone infarction of the metaphysis

**Table 3 ijerph-19-07348-t003:** Steinberg’s classification of avascular necrosis of the femoral head [47].

Stage	Criteria
0	Normal or nondiagnostic radiograph, bone scan, MRI
I	Normal radiographs; abnormal bone scan and/or MRI
	IA—Mild (<15% of femoral head affected)
	IB—Moderate (15% to 30% of femoral head affected)
	IC—Severe (>30% of femoral head affected)
II	Cystic and sclerotic changes in the femoral head
	IIA—Mild (<15% of femoral head affected)
	IIB—Moderate (15% to 30% of femoral head affected)
	IIC—Severe (>30% of femoral head affected)
III	Subchondral collapse (crescent sign) without flattening
	IIIA—Mild (<15% of femoral head affected)
	IIIB—Moderate (15% to 30% of femoral head affected)
	IIIC—Severe (>30% of femoral head affected)
IV	Flattening of femoral head
	IVA—Mild (<15% of femoral head affected)
	IVB—Moderate (15% to 30% of femoral head affected)
	IVC—Severe (>30% of femoral head affected)
V	Joint space narrowing and/or acetabular changes
	VA—Mild
	VB—Moderate
	VC—Severe
VI	Advanced degenerative joint disease

## Data Availability

Not applicable.

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
