# Peer review of "Avascular Necrosis of Femoral Head—Overview and Current State of the Art"

_ijerph, 2022, doi:10.3390/ijerph19127348_

Round 1

Reviewer 1 Report

The article offers a comprehensive overview of the pathophysiology and current concept of clinical management of AVN.

The work reads very well, I found only minor mistakes were/was line 140, unnecessary dot on line 249 formation. [3].

I wish you a lot of success

Author Response

Warmly welcome.
Thank you very much for the review and comments on the publication. Of course, I will take into account all the remarks in the revised version of the article.
I am delighted that there are reviewers who can appreciate the work and write flattering words.
Kind regards
Konarski Wojciech

Reviewer 2 Report

39 - I would say early diagnosis gives physicians options beyond THA. Currently this reads that the physician needs time to think and plan management which is usually not an issue.

51 - Add "adolescent and adult", because infants and children get some blood supply from the artery of the ligamentum teres.

78- reword, using "where" is awkward and colloquial.

88-remove "or hip"

Figure 1 - This case shows a nonunion of the femoral neck due to poor surgical technique. Would omit and include different case.

Figure 2 - 'A' is already showing femoral head collapse and sclerosis. An earlier XR would be better if available, as it appears AVN has already developed prior to the fracture which would make a DHS an inappropriate implant. 

114-119 - This entire paragraph needs to be rewritten if the point was to compare screws to DHS. Cancellous screws aren't even mentioned until the very end.

126 - omit AVN, you have already made clear that is a complication.

128 - add autogenous (many fibular struts used are allogenic)

132- Reference order is incorrect, this 32. 

150 - would not use the term "gamma nailing" as this refers to a single short IMN product made by Stryker and there are others like it with same purpose so would instead say "short intramedullary femoral nail". Targon Femoral neck is unique so more appropriate to use the product name.

158/159 - This is true in the elderly, not young patients who you have already demonstrated get fixed with internal fixation.

183 - lost reduction

184 - there should be further discussion of the difference in blood supply and the presence of the proximal femur physis here. 

201-202 - This paragraph discusses the development of AVN, so a figure of a fracture without AVN has no place here. New image needed.

211 - Metal suppression techniques allow visualization of the bone on MRI even with implants. This should be mentioned and cited.

288 - reword this paragraph. 

Author Response

Warmly welcome.
Thank you very much for the review and comments on the publication. Of course, I will take into account all the remarks in the revised version of the article. Kind regards
Konarski Wojciech

Reviewer 3 Report

The current manuscript 'Avascular necrosis of femoral head – overview and current 2 state of the art' is an extensive review of avascular femoral head necrosis, which is a debilitating condition of the patients femoral bone. The authors present a well-organized study of etiology, diagnosis and treatment of this condition. However, it is unclear what it adds to the field. The recent review by Barney et. al., 2022 has a similar organization and content. The only difference here is the division of the contents into subheading (more organized content) over that review. Apart from this, Matthews et al., 2022 present avascular necrosis seen throughout the body. Several reviews like: George et. al., 2022 and Cao et. al., 2016 also discuss the same. It is unclear as to how this review adds a new perspective to the field. It would help if the authors explained how this review is novel? For instance: are there new treatment strategies presented here that the other reviews did not discuss? Are new conclusions drawn by the authors and novel interpretations given? These points will have to be explained well.

Author Response

Warmly welcome.
Thank you very much for your review of the publication. Of course, I agree with the opinion that similar publications already exist. It is also known that arthroplasty is the gold standard in the treatment of most AVNs of the femoral head. In punblication, I wanted to highlight the problem itself, the main risk factors. In addition, it is difficult to consider treatment options after the fixation of a hip fracture or a trochanteric fracture. When dealing with such cases, we perform arthroplasty in the most majority of cases. Of course, I understand the review, but I think the article is written in an interesting way, giving a certain view that the risk of developing AVN should be considered in any attempt of stabilization in the proximal femur fracture.
Best regards
Wojciech Konarski

Round 2

Reviewer 2 Report

Improved following revisions. Figures better depict AVN. 

Author Response

Thank you very much for your review and reported corrections. Indeed, they add a lot to the publication, and thank you for your comments on the photos. This is how the cooperation between the author and the reviewer should look like. Thanks again for the tips and review.

Reviewer 3 Report

1. The authors are presenting their view on how AVN could develop while trying to stabilize the femur. While a few lines in the text are highlighted, it is better to highlight how this is an extrapolation of what is seen in literature. ie., How is this review coming up with better interpretation of the study results that add more value to the review over the other reviews published in 2022?

2. If the others are missing out on noticing the outcomes in AVN development, it would be good to also argue why this is not seen definitively in the studies.

3. In short, a little more improvement in discussion stating how this review is adding valuable conclusions in the field could be presented in the discussion section. 

Author Response

Dear reviewer,

In response to your comments, we aimed to clarify the role of our paper in the "conclusions" section; nonetheless, we kindly ask for clarification of some of your comments, as we perceive that we are missing the context, which makes them difficult for interpretation and formulating a proper reply. Please see the details below:

1. The authors are presenting their view on how AVN could develop while trying to stabilize the femur. While a few lines in the text are highlighted, it is better to highlight how this is an extrapolation of what is seen in literature. ie., How is this review coming up with better interpretation of the study results that add more value to the review over the other reviews published in 2022?

Thank you for your remark. First, we would avoid classifying our interpretation of data as "better" or "worse" compared to others. Most recently published reviews were systematic, limiting their focus to a particular type of management (i.e., operative management in general or nonoperative management) or population (e.g., pediatric populations). These reviews are of great value to researchers in the area, academics, and some clinicians interested in the particular aspect of AVN. However, our review is narrative in nature and tries to synthesize all aspects of AVN, including the pathophysiology, diagnostic methods, staging, non-operative management options, and surgical management, with some promising techniques under investigation. As clinicians, we see that such reference is of value to other practicing orthopedic surgeons, who seek practical overview and up-to-date knowledge in certain diseases and conditions. Often, they have limited time to review and analyze multiple data sources by themselves.

Nonetheless, we are aware that narrative reviews have some flaws, as the interpretation and selection of data are more subjective than systematic reviews that adhere to a pre-defined protocol. Also, the nature of narrative reviews provides instead a commentary and subjective evaluation of the available data than, for example, meta-analyses based on objective statistical methods. This list is incomplete, of course, but, in our humble opinion, these are the most critical narrative review downsides in the context of our manuscript.

An adequate remark, as requested, was added at the end of the conclusions section.

2. If the others are missing out on noticing the outcomes in AVN development, it would be good to also argue why this is not seen definitively in the studies.

We truly apologize, but we cannot understand this remark. The outcome of AVN development is AVN, but we are convinced that the true meaning was something different; it is also possible that this was a shorthand; please specify what changes we should make in the paper.

3. In short, a little more improvement in discussion stating how this review is adding valuable conclusions in the field could be presented in the discussion section

That was already addressed in the first question - please see our response above. Again, an appropriate disclaimer was added at the end of the conclusions section of the manuscript.